# Choice of the Miniature Inertial Optomechanical Sensor Geometric Parameters with the Help of Their Mechanical Characteristics Modelling

**DOI:** 10.3390/mi14101865

**Published:** 2023-09-28

**Authors:** Lee Kumanchik, Marina Rezinkina, Claus Braxmaier

**Affiliations:** 1Department of Quantum Metrology, Institute for Quantum Technologies, German Aerospace Center (DLR e.V.), 2022 Wilhem-Runge-Straße 10, 89081 Ulm, Germany; lee.kumanchik@dlr.de (L.K.); claus.braxmaier@uni-ulm.de (C.B.); 2Institute for Microelectronics, University of Ulm, 2022 Albert-Einstein-Allee 43, 89081 Ulm, Germany; 3Kharkiv Polytechnic Institute, National Technical University, 2 Kyrpychova Str., 61002 Kharkiv, Ukraine

**Keywords:** mechanical sensors, optomechanical devices, mathematical modelling, mechanical characteristics

## Abstract

In this paper, the mechanical characteristics of a miniature optomechanical accelerometer, similar to those proposed for a wide range of applications, have been investigated. With the help of numerical modelling, characteristics such as eigenfrequencies, quality factor, displacement magnitude, normalized translations, normalized rotations versus eigenfrequencies, as well as spatial distributions of the azimuthal and axial displacements and stored energy density in a wide frequency range starting from the stationary case have been obtained. Dependencies of the main mechanical characteristics versus the minimal and maximal system dimensions have been plotted. Geometries of the optomechanical accelerometers with micron size parts providing the low and the high first eigenfrequencies are presented. It is shown that via the choice of the geometrical parameters, the minimal measured acceleration level can be raised substantially.

## 1. Introduction

Recently developed optomechanical accelerometers possess a number of qualities that determine their wide usage. These qualities include high sensitivity (thermal acceleration noise down to 55 × 10^−11^ m∙s^−2^/√Hz at 1 Hz [1]), immunity to electromagnetic interference [2], and a wide spectral range of measured accelerations, namely from stationary to 15 kHz [3]. Another important quality of such accelerometers is their ability to perform accurate measurements without requiring calibration [3].

The main areas of optomechanical inertial and hybrid quantum inertial sensor applications can be outlined as follows [4]: search for dark matter and dark energy with atom interferometry, the detection of gravitational waves, the monitoring of Earth’s gravitational field over time and its rotation rate in three dimensions, and absolute measurements for metrology. Such sensors are also used in geophysics for monitoring continental uplifts and underground water movements, autonomous operations for space for monitoring large areas during long interrogation times with a very high precision, and for civil engineering applications, namely obtaining gravity signatures.

To appreciate the demanded operational parameters of the optomechanical inertial sensors (OMIS), applications with a wide range of performance parameters should be considered. According to the literature sources, a good candidate is space geodesy missions providing gravity field mapping [5]. Measurements were performed for the very low accelerations frequencies in the diapason from 10^−5^ Hz to 1 Hz. For this, a cold atom accelerometer with a sensitivity of 6 × 10^−10^ m∙s^−2^∙t^−1/2^ (t is the averaging time) was used. There are also described measurements for gravity acceleration with much higher resolution than from the satellite [6]. Measurements were performed in dynamic regime from a ship. Another example of the very low acceleration measurements for gravimetric purposes is described elsewhere [7]. The required characteristics were as follows: vertical accelerations frequency equal to 0.15 Hz, their amplitudes were 1–10 m/s^2^, and precision was below 10^−5^ m/s^2^. Acceleration measurements with the help of the developed mobile atomic gravimeter were used to determine the tidal and seismic waves of distant earthquakes with increased sensitivity. The measured acceleration frequency was in the range of 0.1–10^−6^ Hz, sensitivity was not bigger than 37 μGal/√Hz (1 Gal = 0.01 m/s^2^), and the provided long-term stability was better than 2 μGal. Mobile absolute gravity and density distribution measurements with the help of a portable atom interferometry gravimeter based on two-photon stimulated Raman transitions are described elsewhere [8]. Measured acceleration frequency was close to constant, gain sensitivity equaled to 1.9 mGal/√Hz, and the long-term resolution was 30 μGal. One more area using acceleration measurements is orientation and position measurement [9]. For such investigations, si*x*-axis inertial sensors measuring the linear and angular acceleration were used. Measured acceleration frequency lay in a range between 0.5 Hz and 33 Hz, with 131 LSBs/°/s sensitivity and a full-motion sensing range of ±2000°/s (dps), allowing acceleration measurements within a range of ±16 g [10]. Measurements of the accelerations with bigger frequencies, namely 100–1000 Hz, are necessary for navigational purposes [11,12]. For this, the required positional error, the gyroscope drift, the gyroscope random walk, and other parameters depend on the specific application [13].

Lately, not only classical opto-mechanical accelerometers, but their more advanced versions are used. Such accelerometers are applied to determine the position of the moving vehicles without drift [14]. For this, a stable matter-wave interferometer hybridized with a classical accelerometer was used. The interferometer provides measurements of the accelerations with an up to 400 Hz frequency, with a stability of 10 ng after 11 h of integration. An even higher frequency bandwidth for the measured frequencies is necessary for testing and metrology [15,16], for example, with the three-axis accelerometer in setups for vibration tests of the automotive lidars [13]. The required acceleration frequency range was from 6 Hz up to 2000 Hz and the acceleration range was from 9 m/s^2^ to 12 m/s^2^ in the vertical direction. It is described also the usage of accelerometers for metrology measurements in the frequency range from 0.1 kHz to 15 kHz is also described, as well as acceleration amplitudes varying over two orders of magnitude [16]. It is underlined that due to the fundamental nature of the intrinsic accuracy approach possible with the help of these advanced optomechanical sensors, their area of usage could be extended further, including force and pressure sensors [16].

It is known that the operational frequency range of an accelerometer should be much less than its eigenfrequencies, i.e., the natural frequencies at which a mechanical system is prone to vibrate [1,3]. The aim of this current research is to model the mechanical properties of the proposed miniature optomechanical inertial sensor with micron-sized parts in order to choose its parameters for different operational frequencies corresponding to different areas of application.

## 2. Analytical Determination of the OMIS Mechanical Characteristics

As was mentioned above, optomechanical inertial sensors are used for the high precision measurement of acceleration. The most common ways to improve measurement sensitivity and suppress the thermal noise are to lower the operation temperature or improve the sensor design [17]. In this paper, a new design is analyzed. This design is similar to the design of the sensor consisting of a cylindrical test mass held by a suspension system of the six flexures [18]; this sensor is elaborated to diminish the thermal noise. The OMIS acceleration measurement is based on the usage of a Fabry–Perot optical cavity in which one of the mirrors is located on the test mass of the mechanical oscillator (MO) [1]. The precision and other parameters of an OMIS depend on the MO mechanical characteristics. To simplify the considered parameters, a MO can be presented as a simple harmonic oscillator. The approximate equation for displacement of the MO test mass for constant acceleration application can be written as follows [3]:(1)$$Z=a/\omega_{0}^{2},$$
where *Z* is the test mass displacement, *a* is the acceleration, and *ω*_0_ is the natural frequency of the mechanical oscillator.

The resolution of the acceleration measurement is limited by the test mass thermal fluctuations. When a simple harmonic oscillator is used as a model, minimal measured acceleration above cryogenic temperatures is determined as follows [16,19]:(2)$$a_{\mathrm{th}}=\sqrt{\frac{4k_{B}T\omega_{0}}{mQ}},$$
where *Q* is the mechanical quality factor, *k_B_* = 1.380649 × 10^−23^ J/K is the Boltzmann’s constant, *T* is temperature, and m is the oscillator mass.

From (2) it can be seen how the mass, the quality factor, and the natural frequency qualitatively influence the accuracy of the acceleration measurement. It follows from this formula that to reduce the *a_th_* level, one needs to increase the value of a product *m∙Q* and/or decrease *ω*_0_. Usually, a decrease in *a_th_* is achieved by increasing the *m∙Q* product. Further, it will be shown that by choosing the OMIS geometric parameters, it is possible also to achieve a decrease in *a_th_* by reducing *ω*_0_ without decreasing *m∙Q*.

## 3. Numerical Modelling of OMIS Mechanical Characteristics

Figure 1a shows a photograph of the new miniature OMIS with micron-sized parts and a one euro coin for scale. This geometry will be used in further modelling to analyze the influence of the OMIS parameters on its main characteristics: the eigenfrequencies and the quality factors.

For mathematical modeling of the mechanical processes in OMIS, Comsol Multiphysics 5.6 was used. To consider the processes related to thermo-elasticity, such as the appearance of thermal expansion, the nodes “Viscoelasticity” and “Thermal Expansion” were activated as sub-nodes of the node “Linear elastic material” in Comsol Multiphysics 5.6. To describe mechanical processes in the frequency domain in Comsol Multiphysics, a system of equations of motion with regard to damping is solved (for more details see [20]), considering the fastening of some elements and the character of the applied external influences.

To assess the modelling accuracy, a comparison of the results for increasingly finer meshes was made. It was observed that relative differences between eigenfrequencies calculated with meshes starting from “fine” or even smaller coincided to a difference of less than 1%. As an additional check on the simulation parameters, they were applied to the geometry found in [19] and yielded the same eigenfrequencies as was reported, with a deviation of less than 1%.

Figure 2 shows the calculated distributions of the relative displacement magnitudes within the vectors of the displacement fields (shown by red arrows) for the eigenmodes corresponding to the first eigenfrequency: *f*_1_ = 2.389 kHz (Figure 2a), the second one: *f*_2_ ≈ 4.95∙*f*_1_ = 11.83 kHz (Figure 2b), the third one: *f*_3_ ≈ 8.53∙*f*_1_ = 20.367 kHz (Figure 2c), and the last one: *f*_max_ ≈ 22.2∙*f*_1_ = 53.056 kHz (Figure 2d). These calculations correspond to the OMIS design, which we designate as the “base geometry” (see its photo in Figure 1a). This OMIS has the following parameters: outer diameter *D_out_* = 25 mm; OMIS total thickness *H*_max_ = 3 mm; thickness of the parts connecting the inner cylinder with the fixed constrained outer cylinder (corresponds to the minimal dimension)—spring thickness *d*_min_ = 200 μm; inner diameter of the central aperture *D_in_* =10.4 mm; and diameter of the test mass *d_tm_* = 2.2 mm (designations are shown in Figure 1b).

In order to compare the characteristics of the OMIS with a different design and properties, a parameter such as quality factor (Q) can be used. This parameter cannot be calculated analytically for the geometrically complicated systems, but it can be modelled numerically in Comsol Multiphysics 5.6 considering viscoelasticity, which represents a loss factor damping [20]:*Q* = Im(λ)/2Re(λ),
where λ = −*j*ω is the eigenvalue; *f* = −λ/2π*j* is the eigenfrequency; *j* = (−1)^1/2^; and Re and Im are the real and the imaginary parts of λ that becomes a complex number when the damping is introduced.

It is assumed that the used material possesses isotropic linear elastic properties and the generalized Maxwell model [20] was used for its behaviour description. Because of non-accounting of the other factors influencing the damping and the *Q* level (such as the surrounding medium, temperature, imperfections of the used materials, etc.), the simulated quality factors levels are, as a rule, much less than the experimental levels (see, for instance, review [17]) and cannot be used in the same manner as the absolute values, but only in the same manner as the relative ones.

In the present work, the influence of the geometrical configuration and other parameters on the quality factor was not investigated. However, *Q* accounting is necessary to assess whether this parameter is not reduced for some OMIS. The calculated *Q* levels at different eigenfrequencies enable us to evaluate whether damping at high frequencies is comparable with those at the lower frequencies.

Calculated levels of the quality factors for different eigenfrequencies are shown in Figure 3a by asterisks. Figure 3b,c present calculated levels of the participation factors, namely normalized *Z-* and *X*-translations, which relate to the energy contained within each resonant mode.

As can be seen from the calculations, the quality factor *Q* ~6.3 × 10^6^–7 × 10^6^ is quite high for all eigenfrequencies, and their relative differences do not exceed 11% (see Figure 3a).

The levels of the participation factors characterize the contribution of each eigenfrequency to the corresponding motion. The displacement is written as a linear combination of the eigenmodes [20]:(3)$$u_{k}\approx \sum_{j=1}^{n}q_{k,j}\cdot u_{k,j},$$
where *q_k_*_,*j*_ is (*k*,*j*)-th eigenmode amplitude; *u_k_*_,*j*_ is (*k*,*j*)-th eigenmode shape; *k* corresponds to *x*, *y*, or *z* direction; *j* is the eigenmode number; and *n* is the number of all eigenmodes.

*q_k_*_,*j*_ is written through the eigenmode participation factor Γ*_k_*_,*j*_ as follows [20]:(4)$$q_{k,j}=\frac{S_{a,k}\cdot \Gamma_{k,j}}{\omega_{j}^{2}},$$
where *S_a,k_* is the acceleration spectra in *k*-th direction; ω*_j_* is the *j*-th eigenmode circular frequency; $\Gamma_{k,j}=\phi_{j}^{T}M1_{k}$; ϕ*_j_* is the eigenmode in terms of vector of degrees of freedom; *M* is the mass matrix; and $1_{k}$ is a vector that has the value 1 for the *k*-translation and the value 0 for all other translations.

The level of the *k*-th participation factor *Γ_k,j_* characterizes the degree of influence of the movement in the *k*-th direction on the resulting shape of the movement mode. Thus, knowing the values of the participation factors in different directions, it is possible to predict the dominant directions of movement for different eigenmodes and different frequencies, as well as the dominant modes.

The presented participation factors, namely normalized *Z*- and *X*-translations, were chosen as the most significant parameters for analyzing the influence of the spectrum of eigenfrequencies on the peculiarities of movement. These parameters characterize movement in the direction in which the acceleration is measured (*Z*) and in the direction perpendicular to it (*X* or *Y*). The *X* and *Y* directions for this OMIS design are treated identically. In Figure 3c, eigenfrequencies with different participation factors (for instance, second and third eigenfrequencies) each correspond either to *X* or *Y* directions with almost identical eigenmode shapes.

As analysis has shown for the considered OMIS configuration, levels of such participation factors as normalized *X*, *Y*, *Z* rotations are several orders less than the levels of the normalized translation participation factors. As it follows from the modeling, the largest contribution to the displacement in *Z* direction is associated with the first eigenfrequency (see Figure 2a and Figure 3b) and the second and the third eigenfrequencies are associated with the displacement in the perpendicular directions, namely *X* and *Y* (see Figure 2b,c and Figure 3c). The last eigenfrequency is associated with rotation (see Figure 2d). Nevertheless, values of the participation factors of the normalized *Z*- and *X*-translations can be of the same order (see Figure 3b,c), because the eigenmodes amplitudes are inversely proportional to ω^2^ (see (4)), and the contribution of the second and the third eigenfrequencies relative to the first is less than 1%. Thus, to ensure the absence of the influence of movement in directions perpendicular to the main direction for which the acceleration is measured, it is necessary to increase the ratio between the second and the first eigenfrequencies or to provide a decrease in the translation factors of the perpendicular directions.

Information on the participation factors levels is also necessary to assess the effect of higher harmonics on the system cross-talk. Based on the numerical analysis (see Figure 3b,c) we can conclude that the main type of motion is azimuthal displacement along the *Z*-axis, and the first eigenfrequency provides the main contribution to the resulting motion. Thus, for the proposed OMIS configuration, only the first eigenfrequency can be considered when choosing the operating frequencies.

Another parameter important for characterizing the quality of OMIS is dependence of the azimuthal displacement (*Z*) versus the applied force *F* and correlating it to acceleration *a* (*a* = *F*/*m*, where *m* = 3.1 mg is mass of the all parts of OMIS): *Z* = *f*(*a*). The geometric nonlinearity of the system can cause the nonlinearity of this dependence. To model the processes associated with such non-linearity using Comsol Multiphysics 5.6, the “Non-linearity” node was activated in the “Linear elastic material” node, as well as “Additive strain decomposition” and “Calculate dissipative energy” nodes. In addition, the “Study step” was chosen for this case as the “Stationary” with “Include geometric nonlinearity” option, since in this regime geometric nonlinearity is the most pronounced. To obtain such a dependence, force *F* was applied in the center of the OMIS plane along the *Z* axis. The results of the calculated dependence *Z* = *f*(*a*) are shown in Figure 4. Figure 4a corresponds to a rather broad range of forces and acceleration applications—up to 1000∙*g* (where *g* = 9.81 m/s^2^ is the acceleration due to gravity) to show the degree of nonlinearity. Figure 4b corresponds to the case when accelerations are several times bigger than *g*. In all other cases, the nonlinearity is present, but to a different degree.

Let us consider the OMIS characteristics at a frequency range from *f* = 0.001 Hz up to *f* = 200 Hz, a wideband operational range of the inertial optomechanical accelerometer applications [1,2,3,4,5,6,7]. Simulations of the total displacement (*D*) that practically coincides with the azimuthal displacement (*Z*) and axial displacement (*X*), as well as the stored energy density (*J*) at application of *F_z_* = *m∙g* ≈ 0.03 N, have been performed. Figure 5 presents *D* (a) and *X* (b) displacements as well as *J* (c). In the considered frequency band, all calculated distributions coincide well for all frequencies. Similar calculations carried out for the case of application of the load with a very low frequency, namely 10^−7^ Hz, have shown that the results strongly coincide with the corresponding distributions in the stationary case.

From the comparison of the dependencies, calculated for these very low or stationary frequencies with the distributions shown in Figure 5, it can be concluded that the character of the distributions is almost the same, but all values, namely displacement and stored energy, become approximately 1.4 times bigger.

As can be seen from the comparison of the total displacement amplitude mode (see Figure 5a) and the axial displacement (Figure 5b), their character is different. The azimuthal displacement is concentrated mainly in the zone of the central OMIS part where a mirror is located, and in the radial beams connecting it to the outer fixed constrained contour. Such an azimuthal displacement configuration provides a good reciprocity between the displacement and the applied external acceleration. Quite the opposite, the axial displacement is concentrated in the springs connecting the central test mass with the fixed constrained outer contour. It should be noted that due to the chosen OMIS design, maximal axial displacement is substantially less than the maximal azimuthal one.

## 4. Influence of OMIS Geometrical Parameters on Its Mechanical Characteristics

Here, we simulate the influence of the OMIS geometrical parameters on its mechanical characteristics: eigenfrequencies, quality, and participation factors. It was determined that when the OMIS test mass dimensions are fixed, two geometric parameters, namely thickness of the thinnest detail of OMIS—its springs and total OMIS thickness—have the greatest impact on the eigenfrequency levels.

Modelled distributions for the case of OMIS with springs thicker than for the case considered above, namely *d*_min_ = 500 μm, show that the first eigenfrequency corresponds to *f*_1_ = 8.305 kHz, the second one corresponds to *f*_2_ = 2.05 *f*_1_ = 17.042 kHz, and the third one corresponds to *f*_3_ = 3.15∙*f*_1_ = 26.192 kHz. Figure 6 shows the calculated distributions of the quality factor (Figure 6a) and the participation factor, namely the normalized *Z*-translation (Figure 6b).

Modelled distributions for the case of the OMIS with even thinner springs, *d*_min_ = 50 μm, shows that the first eigenfrequency corresponds to *f*_1_ = 0.318 kHz, the second one corresponds to *f*_2_ ≈ 26.9 *f*_1_ = 8.552 kHz, and the third one corresponds to *f*_3_ ≈ 42.76∙*f*_1_ = 13.598 kHz. Figure 7 shows calculated distributions of the quality factor (Figure 7a) and the participation factor, namely normalized *Z*-translation (Figure 7b) versus eigenfrequencies.

As can be seen from Figure 6a and Figure 7a, the levels and the relative differences of the participation factors for these OMIS configurations possess approximately the same character as for the basic OMIS (see Figure 3a).

To summarize the obtained dependencies of the OMIS characteristics, they were presented as graphs of the first three eigenfrequencies (*f*_1_, *f*_2_, *f*_3_) versus *d*_min_ (Figure 8a,b) and total OMIS thickness *H*_max_ (Figure 8c).

As can be seen from Figure 8a, the bigger *d*_min_—thickness of the springs, the bigger all three first eigenfrequencies. It is connected with an increase in the OMIS stiffness when increasing the spring thickness, as it follows, for instance, from the analytical expression for the first eigenfrequency of a simple resonator [3]:ω_1_ = 2π*f*_1_ = (*k*/*m*)^1/2^,
where *k* is the resonator stiffness and *m* is the resonator mass.

However, the character of the dependencies of the ratios between the second and the first (*f*_2_/*f*_1_) as well as the third and the first (*f*_3_/*f*_1_) eigenfrequencies versus *d*_min_ is not linear (see Figure 8b). Levels of *f*_2_/*f*_1_ and *f*_3_/*f*_1_ are much bigger for the thinner membranes than for the thicker ones. The limit of device thinness is set by the manufacturer’s capabilities.

The influence of *H*_max_ level on the eigenfrequencies values is not as pronounced as the *d*_min_ level influence (see Figure 8c). For the considered parameters, *f*_1_ decreases 1.7 times while the *H*_max_ increases 3 times. Ratios *f*_2_/*f*_1_ remain rather big at all *H*_max_ levels, however, the second and the third eigenfrequencies vary more substantially when *H*_max_ is changed.

Using the dependencies shown in Figure 8 for choice of the OMIS configurations for different purposes, two hypothetically realizable OMIS geometries, which provide the lowest and the highest first eigenfrequency level, have been proposed (see Figure 9).

The mechanical parameters have been calculated for the case of the geometry, providing the low limit of the first eigenfrequency (called “Low eigenfrequency geometry”): *H*_max_ = 5 mm—corresponding to the lower frequency (see Figure 8c), *d*_min_ = 50 μm—corresponding to the lower frequency (see Figure 8a), and *R_in_* = 12.3 mm. Simulation results for OMIS with these parameters are as follows for the eigenfrequencies: *f*_1*l*_ = 45.36 Hz, *f*_2*l*_ ≈ 6.05∙*f*_1*l*_ = 274.47 Hz, and *f*_3*l*_ ≈ 16.61∙*f*_1*l*_ = 753.43 Hz. This OMIS model has rather thin springs as well as thin structures connecting the springs to the fixed constrained outer cylinder (its thickness equals to *R_in_* − *R_out_* = 13.4 − 12.3 = 1.1 mm). The calculated distributions of the quality factors corresponding to such an OMIS geometry versus eigenfrequencies are presented in Figure 9a. Figure 9c presents the mode shape for such an OMIS configuration for the first eigenfrequency.

The geometry, providing the high limit of the first eigenfrequency (called “High eigenfrequency geometry”) has the following parameters: *H*_max_ = 2 mm—corresponding to the higher frequency (see Figure 8c), *d*_min_ = 500 μm—corresponding to the higher frequency (see Figure 8a), and *R_in_* = 10.4 mm. Simulation results for OMIS with these parameters are follows eigenfrequencies: *f*_1*h*_ = 9.91 kHz, *f*_2*h*_ ≈ 1.75∙*f*_1*h*_ = 17.376 kHz, *f*_3*h*_ ≈ 3.18∙*f*_1*h*_ = 31.502 kHz. This OMIS model has thicker springs and bigger thickness of the connecting structures (its thickness equals to *R_in_* − *R_out_* = 13.4 − 10.4 = 3 mm). The distribution of quality factors corresponding to such an OMIS geometry versus eigenfrequencies are presented in Figure 9b. Figure 9d presents the mode shape for such an OMIS configuration for the first eigenfrequency.

As follows from (2), the acceleration noise floor is inversely proportional to the square root of the quality factor and directly proportional to the square root of the angular frequency. As it follows from the obtained results, the relative difference between the square root of the quality factors for the first eigenmode of each case is only 4%. However, the square root of the ratio of the first eigenfrequencies is almost 15. Moreover, the OMIS of the “Low frequency geometry” is 2.5 times thicker and its mass therefore is also bigger. This means that, in accordance with (2), for the geometry corresponding to the case “Low frequency geometry” *a_th_*, the acceleration noise floor can be more than 23 times less than for the case “High frequency geometry”. It can be concluded that with the help of the chosen OMIS parameters, the first eigenfrequencies can be reduced and after that the sensitivity of the acceleration measurements can be raised substantially. However, the bandwidth of the accelerometers determines the necessary value of the first eigenfrequency, which should be at least 10 times bigger. The presented examples demonstrate the range in which first eigenfrequencies and the acceleration measurement accuracy can be chosen.

## 5. Conclusions

Usage of the described mathematical model of the mechanical part of the miniature optomechanical accelerometer with micron size parts enables obtaining such characteristics as eigenfrequency level, quality factor, displacement magnitude, normalized translations, and normalized rotations versus eigenfrequencies. It permits also to evaluate the spatial distributions of the azimuthal, radial and axial displacements, and stored energy density for application of the loads in a rather wide frequency range starting from the stationary case. Analysis of the influence of the geometrical parameters of the optomechanical accelerometer on its main mechanical characteristics has revealed the corresponding dependencies. Such an analysis enabled the design of the OMIS geometrical parameters providing minimal and maximal eigenfrequencies for the parameter range that can be implemented in practice. Further investigations are necessary to provide a more general approach for a broader range of OMIS configurations and their dependence on geometrical parameters, such as OMIS minimal dimension—the spring thickness as well as its maximal axial parameter—the total OMIS thickness. It is necessary also to prolong the experimental investigations of the elaborated OMIS model (see Figure 1a) and to create experimental samples of the proposed OMIS type.

## Figures and Tables

**Figure 1 micromachines-14-01865-f001:**
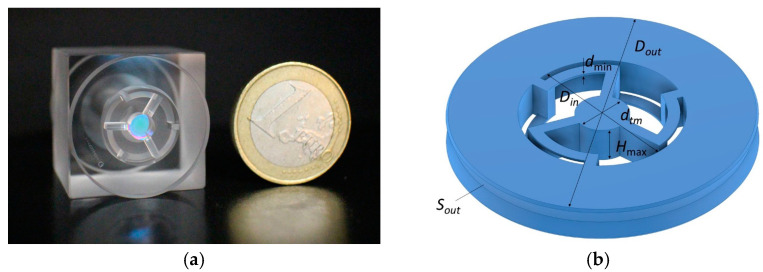
(**a**)—photo of the OMIS model made from one piece of fused silica; (**b**)—OMIS model with its parameter designations (*S_out_* is the OMIS side surface over which it is considered fixed constrained at the mathematical modelling).

**Figure 2 micromachines-14-01865-f002:**
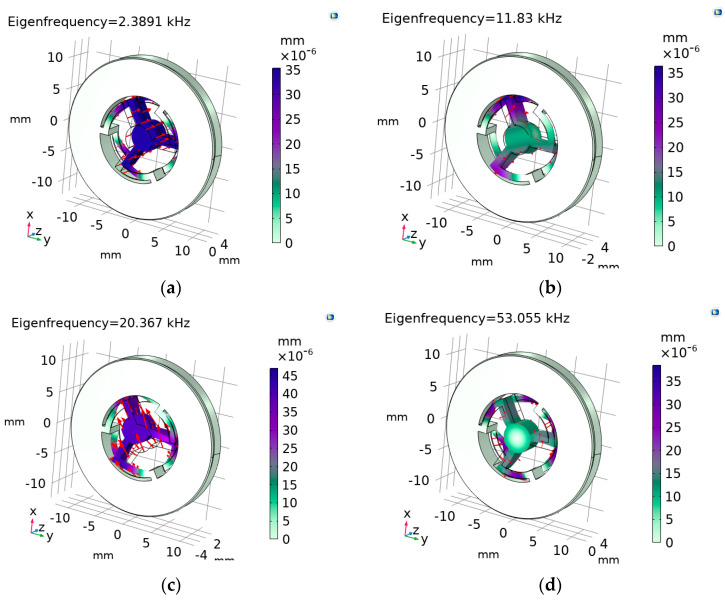
Relative displacement magnitudes for the eigenfrequencies: (**a**–**c**)—the first three, (**d**)—the last one.

**Figure 3 micromachines-14-01865-f003:**
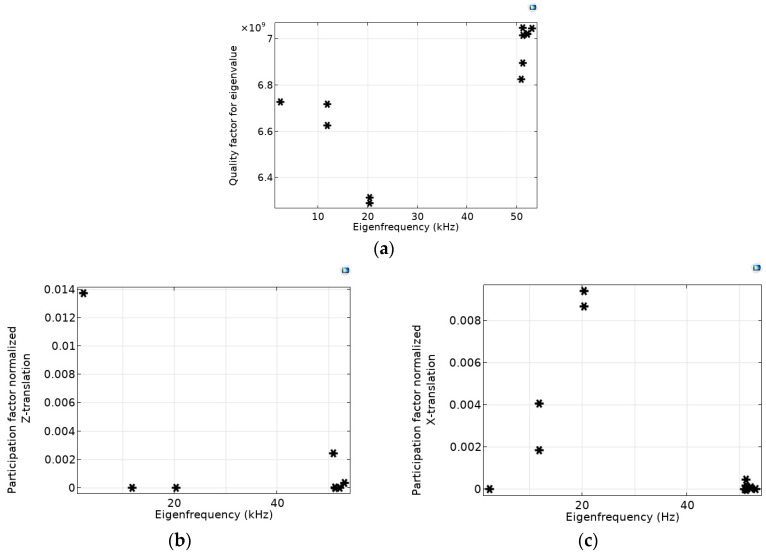
(**a**)—quality factor, (**b**)—normalized *Z*-translation, (**c**)-normalized *X*-translation versus eigenfrequencies (values corresponding different eigenfrequencies are shown by asterisks).

**Figure 4 micromachines-14-01865-f004:**
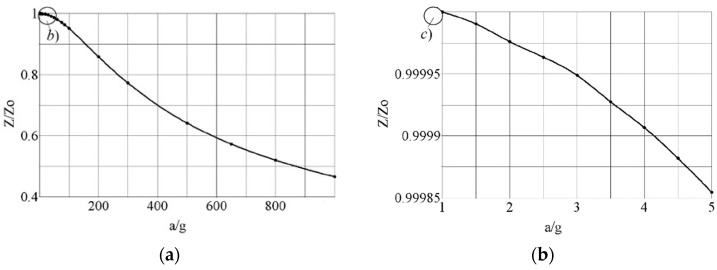
Calculated nonlinear azimuthal displacement (*Z*/*Z*_0_) versus relative applied acceleration *a*/*g*. A stationary load *F_z_* = *m∙g∙k* ≈ 0.03∙*k* N (*k* = 1–1000, *m* = 3.1 mg) is applied in the center of the plane *z* = 0. *Z*_0_ is displacement at *g* acceleration, *Z* is displacement at *a* = *k∙g* acceleration. (**a**)—0 < *a* < 1000∙g; (**b**)—1 < *a* < 5∙g.

**Figure 5 micromachines-14-01865-f005:**
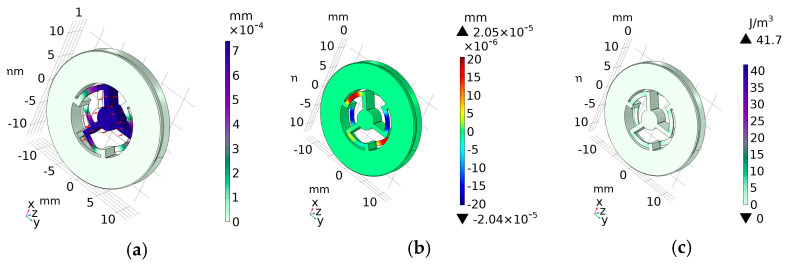
Distributions of the total (**a**) and axial (**b**) displacement as well as stored energy density (**c**) for application of the load *F_z_* = *m∙g* ≈ 0.03 N with frequency 200 Hz in the center of the plane *z* = 0.

**Figure 6 micromachines-14-01865-f006:**
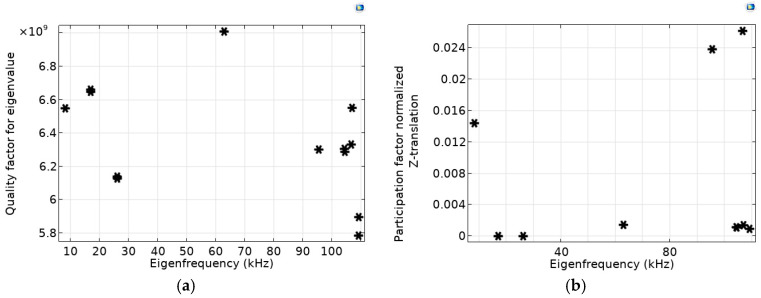
(**a**)—quality factor, (**b**)—normalized *Z*-translation versus eigenfrequencies for OMIS with springs 2.5 times thicker than for the geometry corresponding to Figure 3 (*d*_min_ = 500 μm). Values corresponding different eigenfrequencies are shown by asterisks.

**Figure 7 micromachines-14-01865-f007:**
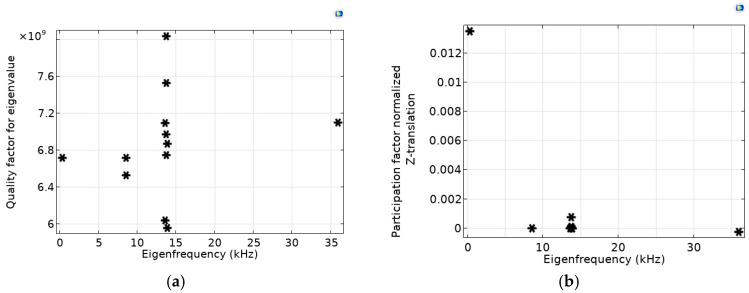
(**a**)—quality factor, (**b**)—normalized *Z*-translation versus eigenfrequencies for OMIS with the springs four times thinner than for the geometry corresponding to Figure 3 (*d*_min_ = 50 μm). Values corresponding different eigenfrequencies are shown by asterisks.

**Figure 8 micromachines-14-01865-f008:**
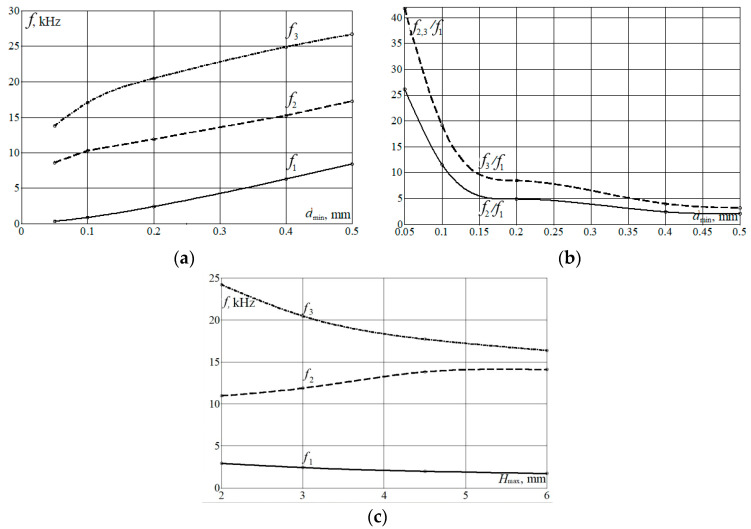
(**a**)—*d*_min_ = var: *f*_3_(*d*_min_), *f*_2_(*d*_min_), *f*_1_(*d*_min_); (**b**) *d*_min_ =var: *f*_2_/*f*_1_(*d*_min_), *f*_3_/*f*_1_(*d*_min_); (**c**) *H*_max_ = var: *f*_3_(*H*_max_), *f*_2_(*H*_max_), *f*_1_(*H*_max_).

**Figure 9 micromachines-14-01865-f009:**
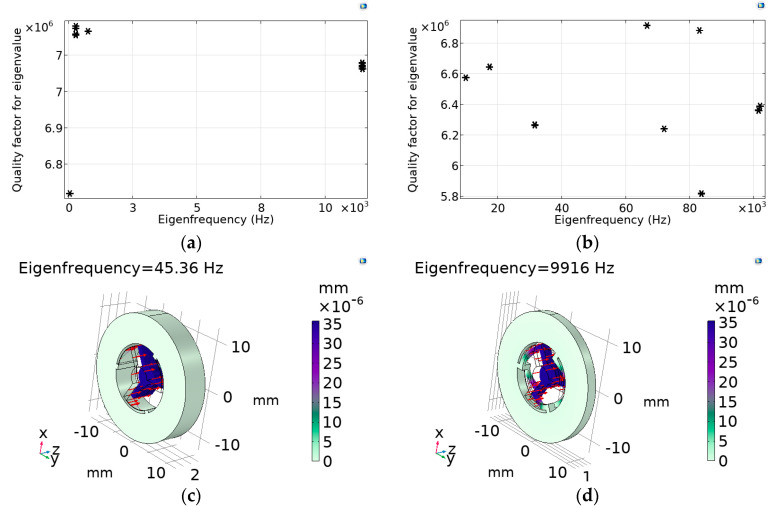
Calculated distributions of the quality factors (**a**,**b**) versus the eigenfrequencies and mode shapes for the first eigenfrequency (**c**,**d**): (**a**,**c**) “Low frequency geometry”; (**b**,**d**) “High frequency geometry” (values corresponding different eigenfrequencies are shown by asterisks).

## Data Availability

The data presented in this study are available upon request from the corresponding author.

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
