# Peer review of "Choice of the Miniature Inertial Optomechanical Sensor Geometric Parameters with the Help of Their Mechanical Characteristics Modelling"

_micromachines, 2023, doi:10.3390/mi14101865_

Round 1

Reviewer 1 Report

In this Manuscript the authors should clearly  explain  what they mean from eigenfrequencies as well as write which equations they are solving particularly. It is unclear throughout the manuscript.

Author Response

Thank you very much for taking the time to review this manuscript. Please find the corresponding corrections below and the corresponding highlighted changes in the re-submitted manuscript.

Comments 1: In this Manuscript the authors should clearly explain what they mean from eigenfrequencies as well as write which equations they are solving particularly. It is unclear throughout the manuscript.

Response: Additional explanations are included (see p. 2, lines 83-85, see p. 3, lines 128-131, marked in yellow).

Reviewer 2 Report

This paper presents a geometric parameter selection method for an optomechanical inertial sensor, which is used to improve the minimum detection level of an accelerometer. However, the author should address some issues that are valuable to the manuscript as a whole, in particular:

1. The author mentions "Quantum Inertial Optomechanical Sensor" in the title, but the article only made some geometric parameter selection for the accelerometer noise, what is the relationship between this and quantum?

2. There are a lot of reports about optomechanical inertial sensors, among which there are a lot of work to suppress the thermal noise limit of acceleration, but the author has not reported enough about the related work in the introduction.

3. The author gives the physical image of the optomechanical inertial sensor in Figure 1. Why not carry out basic property test on it, so that the rationality of the design can be verified by comparing with the simulation results?

4. The author gives the calculation results of quality factors of different frequencies in Figure 3. As we all know, the general quality factors for mechanical resonators have been actually tested. How the author calculated the quality factor through simulation, and whether the influence factors such as self-damping and air damping were considered, please give a detailed explanation of the calculation process.

5. Since the author has provided the physical object of the inertial sensor, why not conduct actual tests to verify whether the actual quality factor is consistent with the simulated quality factor? In addition, the simulation results show that the quality factor is as high as 106, which is difficult to see in ordinary mechanical resonators unless some specific structures are made to inhibit phonon tunneling. What special treatment does the author have in the actual processing?

6. The author obtained the inherent parameters of the resonator (frequency, quality factor, etc.) through simulation calculation, and suggested that the author solve the minimum detection limit according to the theoretical formula of the acceleration thermal noise limit.

No

Author Response

Thank you very much for taking the time to review this manuscript. Please find the corresponding corrections below and the corresponding highlighted changes in the re-submitted manuscript.

Comments 1. The author mentions "Quantum Inertial Optomechanical Sensor" in the title, but the article only made some geometric parameter selection for the accelerometer noise, what is the relationship between this and quantum?

Response. Correction is made. (see a new title in p. 1, line 2)

Comments 2. There are a lot of reports about optomechanical inertial sensors, among which there are a lot of work to suppress the thermal noise limit of acceleration, but the author has not reported enough about the related work in the introduction.

Response. See additional text in p.2 (lines 91-95, marked in light blue) and a new reference [17] (p. 11, line 416-417, marked in light blue)

Comments 3. The author gives the physical image of the optomechanical inertial sensor in Figure 1. Why not carry out basic property test on it, so that the rationality of the design can be verified by comparing with the simulation results?

Response. The aim of the paper is to analyze numerically the properties of the inertial optomechanical sensor (see p. 2, lines 85-88, marked in dark gray), so experimental investigations are out the scope of the presented work. The experimental studies are the theme of the ongoing research.

Additional explanations are included (see p. 11, lines 367-369, marked in light blue).

Comments 4. The author gives the calculation results of quality factors of different frequencies in Figure 3. As we all know, the general quality factors for mechanical resonators have been actually tested. How the author calculated the quality factor through simulation, and whether the influence factors such as self-damping and air damping were considered, please give a detailed explanation of the calculation process.

Response. Additional explanations are included (see p. 4-5, lines 152-171, marked in light blue).

Comments 5. Since the author has provided the physical object of the inertial sensor, why not conduct actual tests to verify whether the actual quality factor is consistent with the simulated quality factor?

Response. The aim of the paper is to analyze numerically the properties of the inertial optomechanical sensor (see p. 2, lines 85-88, marked in dark gray), so experimental investigations are out the scope of the presented work. The experimental studies are the theme of the ongoing research.

Additional explanations are included (see p. 5, lines 161-171; p. 11, lines 367-369, marked in light blue).

In addition, the simulation results show that the quality factor is as high as 106, which is difficult to see in ordinary mechanical resonators unless some specific structures are made to inhibit phonon tunneling. What special treatment does the author have in the actual processing?

Response. Additional explanations are included (see p. 6, lines 161-166, marked in light blue).

Comments 6. The author obtained the inherent parameters of the resonator (frequency, quality factor, etc.) through simulation calculation, and suggested that the author solve the minimum detection limit according to the theoretical formula of the acceleration thermal noise limit.

Response. Additional explanations are included (see p. 3, lines 109-110, marked in light blue and 110-114).

Reviewer 3 Report

The article presents the results of a study of the mechanical characteristics of a miniature optomechanical accelerometer. Using the method of numerical modeling, the dependences of the mechanical characteristics of the accelerometer on the dimensions of the sensitive element were obtained. It is shown that the choice of geometric parameters of the sensitive element affects the minimum measured acceleration level.

The article is interesting and useful for specialists working in the field of optomechanical sensors and can be published.

There are some minor comments.

1. The presented article analyzes the new accelerometer design presented in the J.Carter et al. conference paper and the article contains a link in the text to this conference paper (line 91-92). However, this is not enough. The authors should give (at least in a simplified way) the general design of this accelerometer, this will make it easier for the reader to understand the results obtained. The reference to Carter's conference paper should be moved to the bibliography.

2. Section 4 might better be called “Conclusion”.

Author Response

Thank you very much for taking the time to review this manuscript. Please find the corresponding corrections below and the corresponding highlighted changes in the re-submitted manuscript.

Comments 1. The presented article analyzes the new accelerometer design presented in the J.Carter et al. conference paper and the article contains a link in the text to this conference paper (line 91-92). However, this is not enough. The authors should give (at least in a simplified way) the general design of this accelerometer, this will make it easier for the reader to understand the results obtained. The reference to Carter's conference paper should be moved to the bibliography.

Response. Additional explanations and additional reference are included (see p. 6, lines 93-95; ref. 18, p. 12, lines 418-420, marked in purple).

Comments 2. Section 4 might better be called “Conclusion”.

Response. Corrections are made (see p. 11, line 353, marked in purple).

Reviewer 4 Report

This paper presents a design procedure for optimizing the performance of OMIS based on simulations. The main novelty i found is the use of participation factor in analyzing and optimizing the performance. However, how it is used in the design is not clear. So I have the following questions:

1. regarding participation factor, can you give more details on how it is defined and calculated from the simulation? How is the participation factor related with the device performance? 

2. what does it mean for "Sout is square over which OMIS is considered fixed constrained"? What is Sout?

3. How is Q simulated? Q is only plotted and there is no analysis of it in the text. For example, what we can learn from Q plot, what influences the Q? 

4. In equation 3, what is i?

5. In figure 8, can you add y labels?

6. why for the high frequency structure, we couldn't have 5mm Hmax?

7. "it is necessary to increase the ratio between the first and the second eigenfrequencies". I thought we should decrease instead of increase?

English can be improved. There are some sentences that are too long and complicated, which can be broken down into shorter sentences. 

Author Response

Thank you very much for taking the time to review this manuscript. Please find the corresponding corrections below and the corresponding highlighted changes in the re-submitted manuscript.

Comments 1. regarding participation factor, can you give more details on how it is defined and calculated from the simulation? How is the participation factor related with the device performance?

Response. Additional explanations are included (see p. 6, lines 187-194, marked in green).

Comments 2. what does it mean for "Sout is square over which OMIS is considered fixed constrained"? What is Sout?

Response. Additional explanations are included (see p. 3, lines 122-123, marked in green).

Comments 3. How is Q simulated? Q is only plotted and there is no analysis of it in the text. For example, what we can learn from Q plot, what influences the Q?

Response. Additional explanations are included (see p. 4-5, lines 152-171, marked in light blue; see p. 5, lines 178-179, marked in green; see p. 9, lines 290-292, marked in green; see p. 11, lines 341-342, marked in green).

Comments 4. In equation 3, what is i?

Response. I am sorry, it was a typo (see corrected equation 3, p. 5)

Comments 5. In figure 8, can you add y labels?

Response. Actually, there are labels in figures 8a,c: they are inside each graph in the upper left corner. I have added the label in figure 8b also inside the graph in the upper left corner (see Fig. 8, p. 9).

Comments 6. why for the high frequency structure, we couldn't have 5mm Hmax?

Response. Additional explanations are included (see p. 10, lines 320-321 and 330-332, marked in green).

Comments 7. "it is necessary to increase the ratio between the first and the second eigenfrequencies". I thought we should decrease instead of increase?

Response. I am sorry, it was a typo (see correction in p. 6, line 215)

English can be improved. There are some sentences that are too long and complicated, which can be broken down into shorter sentences.

Response. Corrections are made (see p. 1, lines 33-40, marked in gray; see p. 6, lines 195-199, marked in gray; see p. 7, lines 244-248, marked in gray; see p. 11, lines 354-359, marked in gray; see p. 11, lines 360-367, marked in gray).

Round 2

Reviewer 2 Report

No